# ChronoGAM: An End-to-End One-Class Time Series Gaussian Mixture Model

## Abstract

Recently, several algorithms have been proposed for One Class Learning (OCL) with time series. However, several problems can be found in these methods, problems involving the collapse of hyperspheres, manual thresholds, numerical instabilities and even the use of unlabeled instances during training, which directly violates the concept of OCL. To avoid these problems and solve cases like the numerical instability of some methods this paper proposes an end-to-end method for time series one-class learning based on a Gaussian Mixture Model (GMM). The proposed method combines the unsupervised learning technique of an autoencoder adapted to extract temporal and structural features of a time series, combined with distribution learning, to provide better performance than other state-of-the-art methods for the classification of time series data. ChronoGAM is a novel method that is capable of improving the temporal importance of the representations learned by the autoencoding system. We propose a new objective function with modifications to penalize the small values on the covariance matrix without resulting in exploding gradient propagation, causing numerical instabilities, and adapting the energy calculus to avoid the use of exponential functions. The method is tested on over 85 benchmark datasets, generating 652 datasets. We gain in 369 datasets, with an average ranking of 2.68, being the top-ranked method.

## 1 Introduction

Training machine learning models usually requires many labeled samples belonging to different classes. However, it requires a considerable effort to label each instance. Especially, this effort needs to be performed by domain experts, which makes labeling a longstanding and costly task (Cabral & de Oliveira, 2014; Gao et al., 2020). In some scenarios, there may be interest in identifying instances of a single class, which is commonly easier to obtain data than other classes. For instance, consider the application of quantifying the number of disease-vector mosquitoes in a region using an insect sensor (Reis et al., 2018). In this case, we may consider creating a model to identify the specimens from the *Aedes* or *Anopheles* mosquito genus. If some other insect crosses the sensor, the model may ignore it, independently of its species.

In this scenario, One-Class Learning (OCL) arises as an alternative to the most usual binary or multiclass learning (Moya et al., 1993). OCL focuses on differentiating whether an instance belongs to the interest class by training a model only with the interest instances (Emmert-Streib & Dehmer, 2022). Learning from a single class has been applied to several tasks, such as dealing with severe class imbalance, outlier detection, novelty detection, and concept learning (Alam et al., 2020; Seliya et al., 2021). This paper focuses on OCL for time series. With the increasing ease of access and usage of sensors and other ways to monitor varied phenomena, many applications that use these data have been emerging (Yeh et al., 2018). Similarly, the OCL for time series is gaining more attention in application domains such as health monitoring and fraud detection (Pang et al., 2021; Mou et al., 2023).

Classical algorithms for OCL, such as One-Class Support Vector Machines (Schölkopf et al., 1999), have limitations when applied to time series data due to the difficulties in handling high-dimensional data (Schölkopf et al., 2001). Several other methods have been proposed in the literature to overcome this issue, most relying on deep learning approaches (Pang et al., 2021). Nevertheless, there are still significant gaps in existing techniques and their application to time series data, as well as challenges

related to balancing and stabilizing existing loss functions. Most of these methods are based on learning data representations to address OCL problems using autoencoders (Ruff et al., 2018; Mou et al., 2023; Hayashi et al., 2022; Ren et al., 2019). However, there are two main issues with this premise: (i) the autoencoders used to learn representations are not entirely suitable or adapted for time series, and (ii) the existing methods usually rely on unlabeled instances to learn a robust representation, which directly contradicts the OCL base definition, where only instances of the target class should exist in the training set.

Zong et al. (2018) proposed the Deep Autoencoding Gaussian Mixture Model (DAGMM) to deal with these limitations. DAGMM incorporates errors and similarities in its learned representation while using a network to estimate Gaussian Mixture Model (GMM) parameters and learn the distribution of the components of the class of interest. However, a problem in the objective function used by DAGMM can cause numerical instabilities that lead to model collapse. In this sense, we propose an end-to-end method called Chrono Gaussian Mixture Model (ChronoGAM). We construct our solution based on the assumption that adapting the autoencoder to capture representations considering a time series's temporal and structural features, combined with distribution learning, will perform better than other state-of-the-art methods for time series one-class classification. By regularizing the covariance matrix in the objective function, the GMM will avoid numerical instability that could lead to a learning collapse. In summary, **this paper has the following main contributions**:

1. We propose an end-to-end method capable of outperforming other one-class learning methods by using a Gaussian Mixture Model, improving the temporal dependency on the representations learned by the autoencoder;

2. We balance the loss function used by the DAGMM method to avoid numerical instability in the calculus of the inverse covariance matrix;

3. The proposed method was empirically tested against other state-of-the-art methods for time series one-class learning in over $85$ benchmark datasets using the correct training setups for one-class methods, generating $652$ datasets.

## 2 RELATED WORKS

We divide the related methods into two categories, the two-step methods and end-to-end methods. Each of these categories implements its own methods to define and find interest instances in the data. In the few sections, we define each strategy used by these categories and the problems that each one has when dealing with time series data.

Methods in the Two-Step category use strategies in two different phases. Generally, the first phase uses an autoencoder to learn the data representation based on its reconstruction, while the second phase uses the autoencoder's pre-trained weights to work in a sub-space and map the instances of interest using some strategy. End-to-end methods use only one training phase, some of them work by learning the reconstruction of instances and using the error as a delimiter for the class of interest, while other strategies combine reconstruction with other methods such as learning distributions.

### 2.1 TWO-STEP METHODS

**Deep Support Vector Data Description (DeepSVDD)** (Ruff et al., 2018) uses a deep neural network, typically an autoencoder, to learn a latent representation of the interest class. The network is trained to reconstruct the samples with minimal error. However, instead of using reconstruction error as a direct anomaly measure, DeepSVDD introduces a hypersphere in the learned representation space, aiming to encapsulate the interest instances. During training, the network is optimized to minimize the distance between the embedded representations of the interest instances and the center of the hypersphere. Instances outside the hypersphere are considered as part of a non-interest class. Recently, DeepSVDD-based methods have shown good performance in various domains of one-class learning (Pang et al., 2021). Techniques such as Autoencoding One Class (AOC) (Mou et al., 2023) and Deep Semi-Supervised Anomaly Detection (DeepSAD) (Ruff et al., 2020) are based on the DeepSVDD method. However, these proposed methods use unlabeled instances to learn the initial data representation, which falls into the category of positive-unlabeled learning (PUL) (Bekker & Davis, 2020).

Models can learn representations better if they are trained without unlabeled data. However, this can cause a problem where the representations are too close to the center used in the objective function. This is called a local optimum and was identified by Ruff et al. (2018). The objective function considers the distances from the representations to a center in space. To solve this problem, all representations are mapped to a constant solution equal to the center. This is done during the second-step training, which only considers the labeled instances. This approach tends to approach an optimal solution. However, the model will map all data to this constant solution when new test data is introduced. This results in the classification of all instances as the interest class.

## 2.2 End-to-End Methods

**LSTM-AE** (Provotar et al., 2019) rely on an autoencoder applied to the data to learn the representation of the interest class, and then a threshold is set based on the assumption that by learning the reconstruction of the interest class, non-interest instances will have a higher reconstruction error, and the threshold is used to separate these data (Liao & Yang, 2022; Mujeeb et al., 2019; Provotar et al., 2019). However, these methods face issues such as the sensitivity of the defined threshold, which can increase the false positive rate; dependence on the quality of the reconstruction, where unreliable reconstruction directly impacts the classification of the interest class; and sensitivity to variations in the interest instances, where data deviating from the original distribution may be classified as false negatives.

**Deep AutoEncoder Gaussian Mixture Model (DAGMM)** (Zong et al., 2018) aims to address these issues through a probabilistic modeling approach. Unlike relying solely on reconstruction error, DAGMM utilizes a Gaussian Mixture Model to model the data distribution. The employed AutoEncoder learns a low-dimension latent representation of the data, which is used to estimate the latent density via a Gaussian Mixture Model. Consequently, detecting data outside the target class becomes akin to capturing data outside the distribution. However, the structure proposed for the AutoEncoder is insufficient to capture temporal features or relations on time series data, and the objective function suffers from numerical instability, affecting how the model behaves on different datasets.

## 3 ChronoGAM: Chronology Gaussian Mixture model for time series one-class learning

ChronoGAM combines the unsupervised learning technique of an autoencoder adapted to extract time features with a new Gaussian Mixture Model strategy in an end-to-end way. First, we implement an autoencoder that harnesses the power of convolutions to identify key temporal patterns within the data based on the literature on convolutional approaches for time series classification (Fawaz et al., 2019; Wang et al., 2017). The autoencoder employs a three-block encoder structure, where each block consists of a convolutional layer and an activation. These convolutional layers are instrumental in capturing the underlying structural features of time series (Fawaz et al., 2019). The model comprehensively understands the temporal patterns inherent in the data by extracting these features, which can be interpreted as temporal features since the time series structure represents the time dimension. Furthermore, the inclusion of an activation layer facilitates the reconstruction of the temporal data, ensuring that the model can effectively capture and represent the intricate periodic patterns within the series.

In this aspect, we rely on the snake activation proposed by Ziyin et al. (2020), which has exhibited remarkable performance in reconstructing time series data compared to other common activations used for time series tasks, such as ReLU or LeakyReLU. For the decoder, the layer and block sequence of the encoder are maintained in reverse order and use transposed convolutions instead of normal convolutions. All the kernel sizes, filters, and output dimensions of the layers are preserved in reverse order. This configuration allows the learning of a temporal series reconstruction using the temporal dimension's structural features.

In our Gaussian Mixture Model strategy, similar to the DAGMM (Zong et al., 2018), the objective is to learn a low-dimensional representation considering the features of reconstruction errors and cosine similarity and feed it into the Gaussian Mixture Model. Figure 1 provides an overview of the architecture of the method. Our method learns a low-dimensional representation denoted as $\mathbf{z}_c$

through an encoder. The decoder produces the reconstruction $\hat{\mathbf{x}}$, which is used to calculate the mean squared error (MSE) and cosine similarity, represented by the vector $\mathbf{z}_r$. Together, they form the vector $\mathbf{z}$ and enter the estimation network.

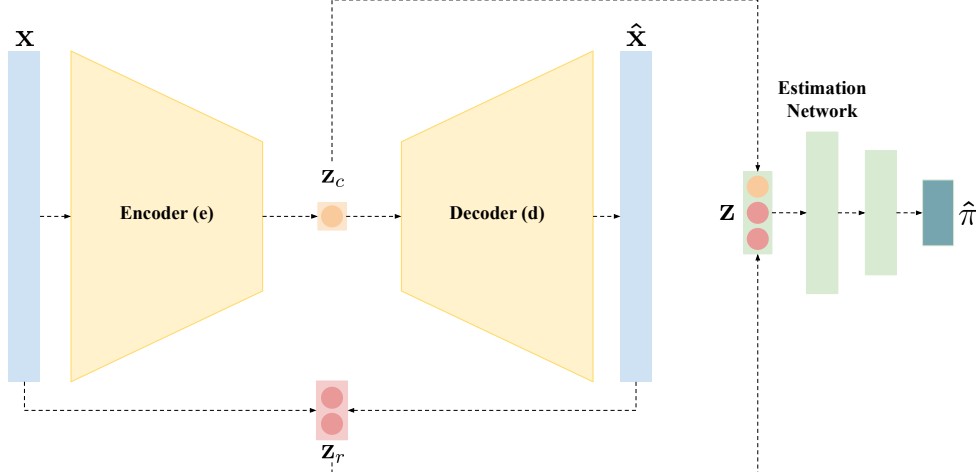

Figure 1: General architecture of ChronoGAM. The structure of the AutoEncoder can be observed, along with how latent representation, reconstruction errors, and cosine similarity (represented by $\mathbf{z}_r$) are used for the GMM parameter estimation network. Adapted from Zong et al. (2018).

Thus, using an encoder function $e$ and a decoder function $d$, the network obtains the values of $\mathbf{z}_c$ and $\hat{\mathbf{x}}$ through the following equations: $\mathbf{z}_c = e(\mathbf{x}; \boldsymbol{\theta}_e)$ and $\hat{\mathbf{x}} = d(\mathbf{z}_c; \boldsymbol{\theta}_d)$. Where $\mathbf{z}_c$ denotes the representation generated by the encoder $e$, $\mathbf{x}$ are the input data, $\hat{\mathbf{x}}$ represents the reconstructed series with the decoder $d$, $\boldsymbol{\theta}_e$ and $\boldsymbol{\theta}_d$ are the encoder and decoder parameters, respectively. To generate the input $\mathbf{z}$ for the estimation network, $\mathbf{z}_r$ is first calculated using the following equation, $\mathbf{z}_r = [\text{mse}(\mathbf{x}, \hat{\mathbf{x}}), s(\mathbf{x}, \hat{\mathbf{x}})]$, where $s(\cdot)$ is the cossine similarity function.

Finally, we can define $\mathbf{z}$ as a multidimensional vector composed by the concatenation of the vectors $\mathbf{z}_c$ and $\mathbf{z}_r$, $\mathbf{z} = [\mathbf{z}_c, \mathbf{z}_r]$. Given the low-dimensional representation, the GMM-based network performs density estimation. During the training phase, using a mixture-component distribution $\phi$, mixture means $\boldsymbol{\mu}$, and mixture covariances $\boldsymbol{\Sigma}$, the estimation network will estimate the GMM parameters and evaluate a modified likelihood/energy for the data. The estimation network consists of an MLP followed by a softmax output, the output of this MLP is the estimation parameters for the GMM components. The density estimation $\hat{\boldsymbol{\pi}}$ can be calculated by $\hat{\boldsymbol{\pi}} = \text{softmax}(\text{MLP}(\mathbf{z}; \boldsymbol{\theta}_m))$, where $\boldsymbol{\theta}_m$ denotes the parameters of the estimation network and $\hat{\boldsymbol{\pi}}$ denotes the soft mixture-component prediction. The estimation network predicts the relevant information for modeling the Gaussian mixture from $\hat{\boldsymbol{\pi}}$. It calculates components such as means, covariance matrices, weights, and energies using the predicted values. The model can estimate the GMM parameters using the following equations:

$$\hat{\phi}_k = \sum_{i=1}^{N} \frac{\hat{\boldsymbol{\pi}}_{ik}}{N}, \qquad \hat{\boldsymbol{\mu}}_k = \frac{\sum_{i=1}^{N} \hat{\boldsymbol{\pi}}_{ik} \mathbf{z}_i}{\sum_{i=1}^{N} \hat{\boldsymbol{\pi}}_{ik}}, \qquad \hat{\boldsymbol{\Sigma}}_k = \frac{\sum_{i=1}^{N} \hat{\boldsymbol{\pi}}_{ik} (\mathbf{z}_i - \hat{\boldsymbol{\mu}}_k)(\mathbf{z}_i - \hat{\boldsymbol{\mu}}_k)^T}{\sum_{i=1}^{N} \hat{\boldsymbol{\pi}}_{ik}}, \quad (1)$$

where $\hat{\phi}_k$ represents the probability of the estimated mixture for the component $k$, $\hat{\boldsymbol{\mu}}_k$ represents the estimated mean for the component $k$, $\hat{\boldsymbol{\Sigma}}$ the mixture covariance for the component $k$, and $k$ is the number of dimensions for $\hat{\boldsymbol{\pi}}$. During the training phase, the threshold for the one-class learning is defined as a percentile of all energies in the training data to avoid the necessity of manually estimating that parameter.

The estimation of the energy of each sample is modified to avoid the use of functions such as $\exp$, which can cause very large values in the numerator of the equation, combined with small values of the covariance matrix, which can cause explosions of gradients in the network, leading to numerical

instabilities and collapse of the model. In this sense, we improve the Energy strategy from Zong et al. (2018) using a variation from the exp function, defined in Equation 2. It only considers the exponential for values below 0, avoiding negative values, and for values greater than 0, the function considers the value itself added to a constant 1 in order to preserve the continuity of the function while preserving the magnitudes.

$$\mathrm{explu}(x) = \begin{cases} \exp(x) & \text{if } x \leq 0, \\ x+1 & \text{otherwise.} \end{cases} \tag{2}$$

Using this $\exp$ variation, the energy estimation equation for each sample is calculated by:

$$E(\mathbf{z}) = -\log \left( \sum_{k=1}^{K} \hat{\phi}_k \frac{\mathrm{explu}(-\frac{1}{2}(\mathbf{z} - \hat{\boldsymbol{\mu}}_k)^T \hat{\boldsymbol{\Sigma}}_k^{-1}(\mathbf{z} - \hat{\boldsymbol{\mu}}_k))}{\sqrt{|2\pi\hat{\boldsymbol{\Sigma}}_k|}} \right). \tag{3}$$

We propose a new objective function with modifications to penalize the small values on the covariance matrix without resulting in exploding gradient propagation, causing numerical instabilities, and adapting the energy calculus to avoid using exponential functions. With this strategy, we reduce the number of cases where the model collapses due to these instabilities in training. The following equation denotes the Objective Function used:

$$\mathcal{L}_t(\boldsymbol{\theta}_e, \boldsymbol{\theta}_d, \boldsymbol{\theta}_m) = \frac{1}{N} \sum_{i=1}^{N} \mathcal{L}_r(\mathbf{x}_i, \hat{\mathbf{x}}_i) + \frac{\lambda_1}{N} \sum_{i=1}^{N} E(\mathbf{z}_i) + \lambda_2 \min \left( \sum_{k=1}^{K} \sum_{j=1}^{d} -\log(\hat{\boldsymbol{\Sigma}}_{kjj}), 100 \right), \tag{4}$$

where $\mathcal{L}_t$ is the total loss, $N$ is the batch size, $\mathcal{L}_r$ is the reconstruction loss, $E$ is the same energy function defined by Equation 3, $\lambda_1$ and $\lambda_2$ are meta-parameters equal to 0.1 and 0.005 respectively.

## 4 EXPERIMENTAL EVALUATION

This section presents the experimental evaluation of this article. We present the used datasets and experimental settings. Our goal is to demonstrate that our ChronoGAM outperforms other state-of-the-art methods. The experimental evaluation codes are publicly available[1].

### 4.1 DATASETS

We evaluated the proposed method on a set of 85 datasets provided by UCR repository (Dau et al., 2018). For the experiments performed, the training and test instances proposed in the data sets were maintained to facilitate the comparison of the results found in the literature. However, to use each dataset in a one-class problem, a transformation is required since only instances of the interest class can be present in the training set.

For each of the 85 datasets collected, each label, or class, is considered as an interest class for the problem. For instance, a dataset with 10 classes will be transformed into 10 different datasets considering each of its labels $(1, \ldots, 10)$ as a class of interest. Thus, after the transformation, only instances of the class of interest are available for training, which results in a smaller training set size, making it more challenging not only because it uses a single class but also because it has few available instances. After the transformation, 652 sets of data are generated, respecting the premise of one-class learning and ready for experiments. The table with the generated datasets and their characteristics can be found in the Github repository since, due to the number of datasets generated, the use of the table in this work would be unfeasible.

---

[1]Ommited link for blind-review purposes.

## 4.2 EXPERIMENTAL SETUP

As classic baselines, we use **OCSVM** Schölkopf et al. (1999) and **IsolationForest** Liu et al. (2008) in order to compare the performance of deep learning algorithms with classic machine learning algorithms for the problem, in this case the entire series are passed to the algorithms. To compare with two-step methods, we use **DeepSVDD** Ruff et al. (2018), which is used as a basis for other algorithms that implement losses based on hyperspheres, and do not use any unlabeled instance for the training phase. As baselines for end-to-end methods, we use the **DAGMM** Zong et al. (2018), which initially would be the main competitor, as it is the basis for this work and do not use any manually defined threshold.

For deep learning baselines, all network characteristics, such as the number of layers, neurons, and activations, are maintained as in the original proposal. With the exception of small adjustments such as the number of neurons in the output layer and activations used in these layers that need to be performed to adapt them to one-class problems with time series. On the ChronoGAM we use three convolutional blocks, the convolutional layers use the respective filter numbers of $(32, 32, 16)$. Each layer employs a kernel of varying sizes, specifically $(7, 5, 3)$, which enables the capture of features in larger temporal contexts, while subsequently narrowing down to extract local features. To produce a one-dimensional representation, a global average pooling and two linear layers with output dimensions of $(64, 1)$ are used. These parameters were chosen through the evaluation of the application of neural networks to capture representations used in classification problems in the literature, as in Fawaz et al. (2019).

## 5 RESULTS

This section discusses the results for each method used as baselines and the ChronoGAM proposed in this work. Altogether, 19560 experiments were carried out with training sets of the datasets, totaling approximately 45007 time series that were evaluated and 1350210 predictions throughout the training phase. The results produce 3912 metrics used to evaluate and generate data to compare the methods in this work.

Following other works from Fawaz et al. (2019); Lucas et al. (2019); Forestier et al. (2017); Petitjean et al. (2016), we use the mean accuracy measure averaged over the experiments on the test set. In our case, we use 5 runs. As Fawaz et al. (2019), we based our analysis in the recomendation of Demšar (2006); Benavoli et al. (2016), using the Friedman test (Friedman, 1940) to reject the null hypothesis and perform a pairwise post-hoc analysis with the average ranking comparinson replaced by the Wilcoxon signed-rank test (Wilcoxon, 1945) with Holm's alpha (5%) correction (Holm, 1979). To facilitate the visualization of the rankings and the comparison between the methods, we rely on the critical difference diagram proposed by Demšar (2006).

### 5.1 UCR DATASETS RESULTS

We provide the analyses and results of the models applied to the 652 datasets generated from the transformation of 85 UCR datasets of univariate time series into one-class problems; the datasets were selected based on works in the literature as in Fawaz et al. (2019). The critical difference diagram corresponding to the tests performed with the 652 datasets is shown in Figure 2. It is possible to observe that ChronoGAM using the strategy of regularizing the covariance matrix, outperforms the other algorithms with an average ranking of approximately 2.25, while without using the regularizer (wo. Reg), it is in second place with an average ranking of 2.68. ChronoGAM wins in 369 problems out of the 652 generated, significantly outperforming the other algorithms and their two main competing baselines, DAGMM and IsolationForest. Table 1 shows the count of wins for each algorithm.

ChronoGAM has been more successful than its main competitor (DAGMM) for two primary reasons. Firstly, its autoencoder architecture is more flexible, using convolutional structures that are able to detect temporal correlations in the data, as demonstrated in various studies on time series classification (Fawaz et al., 2019; Wang et al., 2017; Zong et al., 2018). In contrast, DAGMM only uses an MLP-based autoencoder, which cannot capture these dependencies. The second reason is the covariance matrix regularizer, while the DAGMM uses a $p = \sum_{k=1}^{K} \sum_{j=1}^{d} \frac{1}{\hat{\Sigma}_{kjj}}$

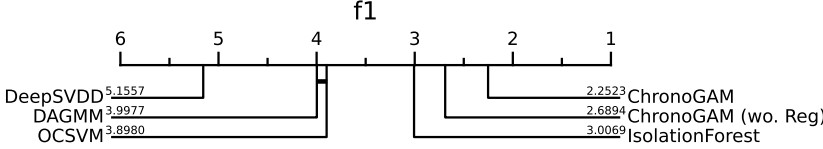

Figure 2: Critical difference diagram showing pairwise statistical difference comparison of 7 one-class methods on the 652 generated datasets from 85 UCR time series classification repository.

| Method | Number of Top One |
|---|:---:|
| **ChronoGAM** | **369** |
| IsolationForest (Liu et al., 2008) | 163 |
| DAGMM (Zong et al., 2018) | 72 |
| OCSVM (Chen et al., 2001) | 44 |
| DeepSVDD (Ruff et al., 2018) | 4 |

Table 1: Table comparing the win count of each method in each of the 652 datasets generated from the 85 selected from the UCR repository.

which can easily cause very large values, making it difficult to correctly propagate the error to the task, leading to numerical instabilities and exploding gradients, ChronoGAM uses the penalty $p = \min(\sum_{k=1}^{K} \sum_{j=1}^{d} -\log(\hat{\Sigma}_{kjj}), \alpha)$ where $\alpha = 100$ , so values that would cause the gradient to explode and cause numerical instabilities as well as disregarding the other tasks that make up the objective function to simply minimize the covariance matrix are more easily avoided.

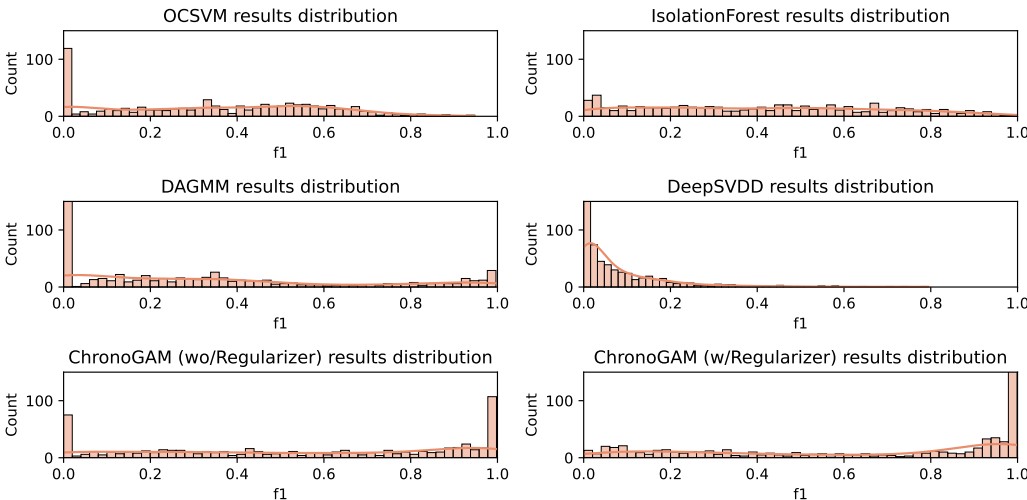

Figure 3: Distribution of the f1 generated from the results of each of the algorithms tested in the 652 data sets generated from the 85 in the UCR repository.

Figure 3 shows the distribution of the count of the f1 score ranges obtained by each of the algorithms. The more counts to the right of the distributions, the better the result of the model. It is possible to notice models susceptible to collapses or failures. In these cases, a number of f1 scores obtained in the range closest to 0 are observed. This behavior is common in models such as the DAGMM, which has numerical instability due to the covariance matrix, and the DeepSVDD, which suffers from hypersphere collapse. Models like IsolationForest distributes their results along the range of possible values for an f1, obtaining a good average ranking. However, it is noted that good results are rare and almost uncommon. ChronoGAM manages to better distribute these results in which the performance was previously average at the end of the range of possible values, achieving state-of-the-art results and avoiding numerical instability with regularization as in the DAGMM.

Note that ChronoGAM has a better average ranking as shown in the critical difference diagram in Figure 2, Figure 3 shows that the distribution of ChronoGAM's f1 is more similar to the objective, which is a higher count of values close to 1. The Table 1 shows the count of results in which each algorithm won in the 652 sets tested. ChronoGAM also maintains a higher count when compared to its competitors. Thus, the results obtained demonstrate that ChronoGAM outperforms other state-of-the-art methods.

## 5.2 WHAT WE CAN EXTRACT FROM THE RESULTS?

Within the UCR repository, each dataset has a domain classification, i.e., a theme. Table 2 compares algorithms in each of the themes in the UCR dataset repository. It is possible to notice that there is again a dominance of ChronoGAM across different scenarios. With the exception of only two cases. The first one concerns the cases of data sets with the Device theme, in which IsolationForest beats ChronoGAM with a good difference in performance. However, due to the sample of 6 sets, it is not possible to say based on this that IsolationForest will always beat ChronoGAM. The second theme is Spectro, composed of 7 data sets. IsolationForest surpasses the performance of ChronoGAM minimally, but it is possible to consider that due to the small difference between the performances, with the correct adjustment for the appropriate data sets, ChronoGAM surpasses the performance of IsolationForest.

| Theme (#) | OCSVM | IsolationForest | DeepSVDD | DAGMM | ChronoGAM |
|---|---|---|---|---|---|
| Image (29) | 0.33 | 0.36 | 0.07 | 0.35 | **0.66** |
| Sensor (18) | 0.25 | 0.36 | 0.10 | 0.25 | **0.57** |
| Motion (14) | 0.32 | 0.42 | 0.09 | 0.29 | **0.50** |
| Device (6) | 0.19 | **0.45** | 0.11 | 0.13 | 0.10 |
| Spectro (7) | 0.48 | **0.58** | 0.18 | 0.31 | 0.57 |
| Simulated (5) | 0.31 | 0.48 | 0.09 | 0.25 | **0.61** |
| ECG (6) | 0.54 | 0.54 | 0.03 | 0.46 | **0.89** |

Table 2: Algorithms' performance grouped by theme. Each entry presents the mean f1 score for the algorithm in that theme. Bold indicates the best results.

Another characteristic of the data that is possible to observe is the size of the time series in each of the data sets. Small considerations need to be taken when using a neural network model to extract features from time series. Table 3 shows the algorithms' results on the sets within each delimited size range. Again, it is possible to notice that ChronoGAM outperforms the other methods in most cases, with the exception of cases where the time series sizes are very small. In these cases, the size of the series impacts the performance of the algorithm due to the size of the kernels used by ChronoGAM, which in very small series may not capture local patterns in the data since the size of the kernels becomes very large. Again, due to the small number of datasets with array sizes less than 81, precisely 14 sets, it is not possible to state that IsolationForest will outperform ChronoGAM on the task.

| Length (#) | OCSVM | IsolationForest | DeepSVDD | DAGMM | ChronoGAM |
|---|---|---|---|---|---|
| <81 (14) | 0.40 | **0.56** | 0.22 | 0.20 | 0.44 |
| 81-250 (19) | 0.39 | 0.41 | 0.06 | 0.33 | **0.62** |
| 251-450 (22) | 0.28 | 0.34 | 0.08 | 0.34 | **0.64** |
| 451-700 (13) | 0.37 | 0.46 | 0.07 | 0.37 | **0.64** |
| 701-1000 (10) | 0.49 | 0.55 | 0.03 | 0.43 | **0.79** |
| >1000 (7) | 0.14 | 0.21 | 0.08 | 0.17 | **0.51** |

Table 3: Algorithm's performance by time series length. Each entry presents the mean f1 score for the algorithm in that length range. Bold indicates the best model.

Figure 4 shows the representations learned by comparing DeepSVDD's hypersphere strategy with ChronoGAM. We observe the collapse of DeepSVDD mentioned in Section 2.1, where the model maps all instances into the hypersphere since training only uses interest instances. We highlight the useful representations learned by ChronoGAM for the one-class time series classification since we

can discriminate the classes by learning from only one class. The representations of the figure were generated from the test sets, i.e., from instances not used in the model's training.

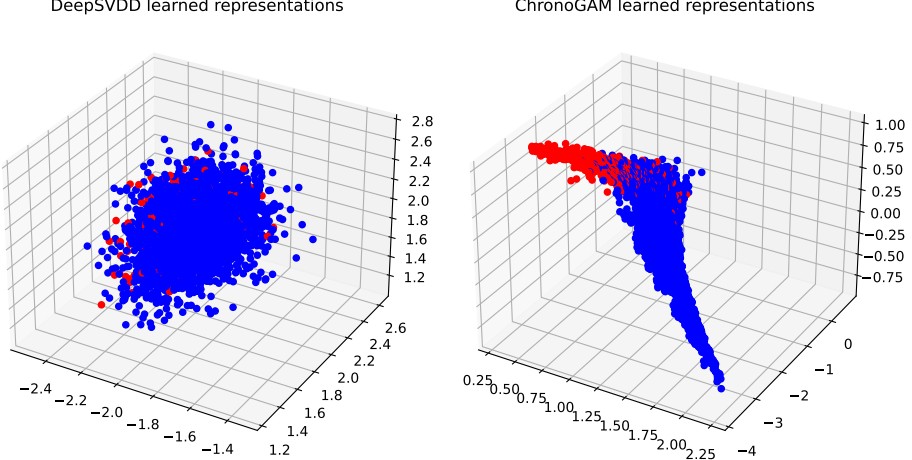

Figure 4: Representations learned during training of the DeepSVDD and ChronoGAM methods in three dimensions on the UWaveGestureLibraryAll dataset (interest class: 1). It is possible to observe the collapse of DeepSVDD on the right, where instances of the class of interest (in blue) and those of non-interest (in red) are mapped together close to the average of the representations.

Finally, it should be noted that our results showed that ChronoGAM outperforms other state-of-the-art methods based on classic methods, such as IsolationForest, and based on deep learning, such as DAGMM. Tests on $652$ datasets generated by transforming the $85$ UCR sets to one-class classification problem sets showed that ChronoGAM has the best average ranking. The count of results and an investigation of these counts' distributions showed that ChronoGAM successfully avoided cases where the other algorithms collapsed or presented instabilities. The average per theme shows that with the exception of specific cases where the performance of IsolationForest can benefit from the nature of the data, ChronoGAM loses in performance, but in $5$ out of $7$ themes, it outperforms the other algorithms. We also show that, due to some concepts of the convolutions, such as the size of the kernels, the model can be impacted by the size of the series. However, in the $6$ range of analyzed values, ChronoGAM surpasses its competitors in $5$ of them.

## 6 CONCLUSION

In this paper, we have proposed the ChronoGAM method, a deep learning method that uses a Gaussian Mixture Model (GMM) strategy with an autoencoder capable of capturing time dependency on the data for time series one-class learning. The new objective function is capable of avoiding numerical instabilities and also penalizing small values on the covariance matrix, which can cause problems on the GMM, using Equation 4, so the model will avoid small $\hat{\Sigma}$ values that will increase the loss cost. We discuss some directions for future work and weaknesses: **1)** Apply cross-dimension dependency for multivariate time series since normal convolutions cannot be the best way to capture the relation between time and structure in multivariate time series. **2)** Use a Normalizing Flow to learn the transformation of a complex data distribution on a Gaussian distribution, the GMM supposes that the data are represented by a Gaussian distribution, which can cause poor performance for some problems. **3)** We discuss the weaknesses of other works, showing that for some other methods, we have problems of constant solutions; in future works, we leave the exciting research idea of mapping the representation learned by the interest class of more flexible manners, i.e., learning distributions or other subspaces that are not a hypersphere.

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

## A  DETAIL OF EXPERIMETS

### A.1  BENCHMARK DATASETS

As in Fawaz et al. (2019) we conduct the experiments on the following UCR benchmark datasets:

**1) Adiac** series have length of 176, 37 classes, and original train/test splits are 390/391. This dataset is part of the Image theme datasets.
**2) ArrowHead** series have length of 251, 3 classes, and original train/test splits are 36/175. This dataset is part of the Image theme datasets.
**3) Beef** series have length of 470, 5 classes, and original train/test splits are 30/30. This dataset is part of the Spectro theme datasets.
**4) BeetleFly** series have length of 512, 2 classes, and original train/test splits are 20/20. This dataset is part of the Image theme datasets.
**5) BirdChicken** series have length of 512, 2 classes, and original train/test splits are 20/20. This dataset is part of the Image theme datasets.
**6) Car** series have length of 577, 4 classes, and original train/test splits are 60/60. This dataset is part of the Sensor theme datasets.
**7) CBF** series have length of 128, 3 classes, and original train/test splits are 30/900. This dataset is part of the Simulated theme datasets.
**8) ChlorineConcentration** series have length of 166, 3 classes, and original train/test splits are 467/3840. This dataset is part of the Sensor theme datasets.
**9) CinCECGTorso** series have length of 1639, 4 classes, and original train/test splits are 40/1380. This dataset is part of the Sensor theme datasets.
**10) Coffee** series have length of 286, 2 classes, and original train/test splits are 28/28. This dataset is part of the Spectro theme datasets.
**11) Computers** series have length of 720, 2 classes, and original train/test splits are 250/250. This dataset is part of the Device theme datasets.
**12) CricketX** series have length of 300, 12 classes, and original train/test splits are 390/390. This dataset is part of the Motion theme datasets.
**13) CricketY** series have length of 300, 12 classes, and original train/test splits are 390/390. This dataset is part of the Motion theme datasets.
**14) CricketZ** series have length of 300, 12 classes, and original train/test splits are 390/390. This dataset is part of the Motion theme datasets.
**15) DiatomSizeReduction** series have length of 345, 4 classes, and original train/test splits are 16/306. This dataset is part of the Image theme datasets.
**16) DistalPhalanxOutlineAgeGroup** series have length of 80, 3 classes, and original train/test splits are 400/139. This dataset is part of the Image theme datasets.
**17) DistalPhalanxOutlineCorrect** series have length of 80, 2 classes, and original train/test splits

are 600/276. This dataset is part of the Image theme datasets.

**18) DistalPhalanxTW** series have length of 80, 6 classes, and original train/test splits are 400/139. This dataset is part of the Image theme datasets.

**19) Earthquakes** series have length of 512, 2 classes, and original train/test splits are 322/139. This dataset is part of the Sensor theme datasets.

**20) ECG200** series have length of 96, 2 classes, and original train/test splits are 100/100. This dataset is part of the ECG theme datasets.

**21) ECG5000** series have length of 140, 5 classes, and original train/test splits are 500/4500. This dataset is part of the ECG theme datasets.

**22) ECGFiveDays** series have length of 136, 2 classes, and original train/test splits are 23/861. This dataset is part of the ECG theme datasets.

**23) ElectricDevices** series have length of 96, 7 classes, and original train/test splits are 8926/7711. This dataset is part of the Device theme datasets.

**24) FaceAll** series have length of 131, 14 classes, and original train/test splits are 560/1690. This dataset is part of the Image theme datasets.

**25) FaceFour** series have length of 350, 4 classes, and original train/test splits are 24/88. This dataset is part of the Image theme datasets.

**26) FacesUCR** series have length of 131, 14 classes, and original train/test splits are 200/2050. This dataset is part of the Image theme datasets.

**27) FiftyWords** series have length of 270, 50 classes, and original train/test splits are 450/455. This dataset is part of the Image theme datasets.

**28) Fish** series have length of 463, 7 classes, and original train/test splits are 175/175. This dataset is part of the Image theme datasets.

**29) FordA** series have length of 500, 2 classes, and original train/test splits are 3601/1320. This dataset is part of the Sensor theme datasets.

**30) FordB** series have length of 500, 2 classes, and original train/test splits are 3636/810. This dataset is part of the Sensor theme datasets.

**31) GunPoint** series have length of 150, 2 classes, and original train/test splits are 50/150. This dataset is part of the Motion theme datasets.

**32) Ham** series have length of 431, 2 classes, and original train/test splits are 109/105. This dataset is part of the Spectro theme datasets.

**33) HandOutlines** series have length of 2709, 2 classes, and original train/test splits are 1000/370. This dataset is part of the Image theme datasets.

**34) Haptics** series have length of 1092, 5 classes, and original train/test splits are 155/308. This dataset is part of the Motion theme datasets.

**35) Herring** series have length of 512, 2 classes, and original train/test splits are 64/64. This dataset is part of the Image theme datasets.

**36) InlineSkate** series have length of 1882, 7 classes, and original train/test splits are 100/550. This dataset is part of the Motion theme datasets.

**37) InsectWingbeatSound** series have length of 256, 11 classes, and original train/test splits are 220/1980. This dataset is part of the Sensor theme datasets.

**38) ItalyPowerDemand** series have length of 24, 2 classes, and original train/test splits are 67/1029. This dataset is part of the Sensor theme datasets.

**39) LargeKitchenAppliances** series have length of 720, 3 classes, and original train/test splits are 375/375. This dataset is part of the Device theme datasets.

**40) Lightning2** series have length of 637, 2 classes, and original train/test splits are 60/61. This dataset is part of the Sensor theme datasets.

**41) Lightning7** series have length of 319, 7 classes, and original train/test splits are 70/73. This dataset is part of the Sensor theme datasets.

**42) Mallat** series have length of 1024, 8 classes, and original train/test splits are 55/2345. This dataset is part of the Simulated theme datasets.

**43) Meat** series have length of 448, 3 classes, and original train/test splits are 60/60. This dataset is part of the Spectro theme datasets.

**44) MedicalImages** series have length of 99, 10 classes, and original train/test splits are 381/760. This dataset is part of the Image theme datasets.

**45) MiddlePhalanxOutlineAgeGroup** series have length of 80, 3 classes, and original train/test splits are 400/154. This dataset is part of the Image theme datasets.

**46) MiddlePhalanxOutlineCorrect** series have length of 80, 2 classes, and original train/test splits are 600/291. This dataset is part of the Image theme datasets.

**47) MiddlePhalanxTW** series have length of 80, 6 classes, and original train/test splits are 399/154. This dataset is part of the Image theme datasets.

**48) MoteStrain** series have length of 84, 2 classes, and original train/test splits are 20/1252. This dataset is part of the Sensor theme datasets.

**49) NonInvasiveFetalECGThorax1** series have length of 750, 42 classes, and original train/test splits are 1800/1965. This dataset is part of the ECG theme datasets.

**50) NonInvasiveFetalECGThorax2** series have length of 750, 42 classes, and original train/test splits are 1800/1965. This dataset is part of the ECG theme datasets.

**51) OliveOil** series have length of 570, 4 classes, and original train/test splits are 30/30. This dataset is part of the Spectro theme datasets.

**52) OSULeaf** series have length of 427, 6 classes, and original train/test splits are 200/242. This dataset is part of the Image theme datasets.

**53) PhalangesOutlinesCorrect** series have length of 80, 2 classes, and original train/test splits are 1800/858. This dataset is part of the Image theme datasets.

**54) Phoneme** series have length of 1024, 39 classes, and original train/test splits are 214/1896. This dataset is part of the Sensor theme datasets.

**55) Plane** series have length of 144, 7 classes, and original train/test splits are 105/105. This dataset is part of the Sensor theme datasets.

**56) ProximalPhalanxOutlineAgeGroup** series have length of 80, 3 classes, and original train/test splits are 400/205. This dataset is part of the Image theme datasets.

**57) ProximalPhalanxOutlineCorrect** series have length of 80, 2 classes, and original train/test splits are 600/291. This dataset is part of the Image theme datasets.

**58) ProximalPhalanxTW** series have length of 80, 6 classes, and original train/test splits are 400/205. This dataset is part of the Image theme datasets.

**59) RefrigerationDevices** series have length of 720, 3 classes, and original train/test splits are 375/375. This dataset is part of the Device theme datasets.

**60) ScreenType** series have length of 720, 3 classes, and original train/test splits are 375/375. This dataset is part of the Device theme datasets.

**61) ShapeletSim** series have length of 500, 2 classes, and original train/test splits are 20/180. This dataset is part of the Simulated theme datasets.

**62) ShapesAll** series have length of 512, 60 classes, and original train/test splits are 600/600. This dataset is part of the Image theme datasets.

**63) SmallKitchenAppliances** series have length of 720, 3 classes, and original train/test splits are 375/375. This dataset is part of the Device theme datasets.

**64) SonyAIBORobotSurface1** series have length of 70, 2 classes, and original train/test splits are 20/601. This dataset is part of the Sensor theme datasets.

**65) SonyAIBORobotSurface2** series have length of 65, 2 classes, and original train/test splits are 27/953. This dataset is part of the Sensor theme datasets.

**66) StarLightCurves** series have length of 1024, 3 classes, and original train/test splits are 1000/8236. This dataset is part of the Sensor theme datasets.

**67) Strawberry** series have length of 235, 2 classes, and original train/test splits are 613/370. This dataset is part of the Spectro theme datasets.

**68) SwedishLeaf** series have length of 128, 15 classes, and original train/test splits are 500/625. This dataset is part of the Image theme datasets.

**69) Symbols** series have length of 398, 6 classes, and original train/test splits are 25/995. This dataset is part of the Image theme datasets.

**70) SyntheticControl** series have length of 60, 6 classes, and original train/test splits are 300/300. This dataset is part of the Simulated theme datasets.

**71) ToeSegmentation1** series have length of 277, 2 classes, and original train/test splits are 40/228. This dataset is part of the Motion theme datasets.

**72) ToeSegmentation2** series have length of 343, 2 classes, and original train/test splits are 36/130. This dataset is part of the Motion theme datasets.

**73) Trace** series have length of 275, 4 classes, and original train/test splits are 100/100. This dataset is part of the Sensor theme datasets.

**74) TwoLeadECG** series have length of 82, 2 classes, and original train/test splits are 23/1139. This dataset is part of the ECG theme datasets.

**75) TwoPatterns** series have length of 128, 4 classes, and original train/test splits are 1000/4000. This dataset is part of the Simulated theme datasets.

**76) UWaveGestureLibraryAll** series have length of 945, 8 classes, and original train/test splits are

896/3582. This dataset is part of the Motion theme datasets.

**77) UWaveGestureLibraryX** series have length of 315, 8 classes, and original train/test splits are 896/3582. This dataset is part of the Motion theme datasets.

**78) UWaveGestureLibraryY** series have length of 315, 8 classes, and original train/test splits are 896/3582. This dataset is part of the Motion theme datasets.

**79) UWaveGestureLibraryZ** series have length of 315, 8 classes, and original train/test splits are 896/3582. This dataset is part of the Motion theme datasets.

**80) Wafer** series have length of 152, 2 classes, and original train/test splits are 1000/6164. This dataset is part of the Sensor theme datasets.

**81) Wine** series have length of 234, 2 classes, and original train/test splits are 57/54. This dataset is part of the Spectro theme datasets.

**82) WordSynonyms** series have length of 270, 25 classes, and original train/test splits are 267/638. This dataset is part of the Image theme datasets.

**83) Worms** series have length of 900, 5 classes, and original train/test splits are 181/77. This dataset is part of the Motion theme datasets.

**84) WormsTwoClass** series have length of 900, 2 classes, and original train/test splits are 181/77. This dataset is part of the Motion theme datasets.

**85) Yoga** series have length of 426, 2 classes, and original train/test splits are 300/3000. This dataset is part of the Image theme datasets.

## A.2 MODEL HYPERPARAMETERS

For each of the baselines, the following hyperparameters were used:

**1) OCSVM:** kernel={linear, poly, rbf}, degree={2, 3}, $\gamma$={scale, auto}, $\nu$={0.5, 0.6, 0.7, 0.8, 0.9, 1.}.

**2) IsolationForest:** n_estimators={100, 200, 300}, max_samples=auto, contamination=auto, max_features=1.

**3) DeepSVDD:** epoch=350, lr={$10^{-3}$, $10^{-4}$, $10^{-5}$, $10^{-6}$}, weight_decay={$10^{-5}$, $10^{-6}$}, optimizer=Adam, activation=LeakyReLU, filters=(128, 256, 128, 32), kernels=(7, 5, 3), representation_dim={16, 32, 64} (The inverse filters and kernels are used in the decoder module).

**4) DAGMM:** epoch=350, lr={$10^{-3}$, $10^{-4}$, $10^{-5}$, $10^{-6}$}, lr_epochs_step=(200, 215, 230), lr_decay={0, 0.1} optimizer=Adam, activation=Tanh, features=(120, 60, 30), representation_dim={2, 3}, estimation_features=({2, 3}, 10), dropout=0.3, 0.4, 0.5 (The inverse features are used in the decoder module).

**5) ChronoGAM:** epoch=350, lr={$10^{-3}$, $10^{-4}$, $10^{-5}$, $10^{-6}$}, lr_epochs_step=(200, 215, 230), lr_decay={0, 0.1} optimizer=Adam, activation={LeakyReLU, Tanh, Snake}, filters=(16, 32, 64), kernels=(7, 5, 3), representation_dim={2, 3}, estimation_features=({2, 3}, 32), dropout=0.3, 0.4, 0.5 (The inverse filters and kernels are used in the decoder module).

All models and baselines are implemented in PyTorch and trained on a single NVIDIA RTX 3060 Ti with 16GB memory.

# B EXTRA RESULTS

## B.1 SHOWCASE OF MAIN RESULTS

Figure 5 shows some more representations generated from datasets used in this work. Each row represents a data set while each column represents a model. It is possible to observe that in all cases for the representations generated with DeepSVDD the fact mentioned in Section 2.1 occurs, where the model finds a constant solution for all instances of the set, mapping all in a hyper-sphere format when trying to encompass them. For the FordA set, considering the class of interest as label 1, DAGMM found instabilities such as those mentioned previously. In this way, the model maps all instances as values close to infinity, generating the visualization of a single point due to the non-numeric values caused by the explosion of the gradients.

It is evident that there is a distinction between the representations produced by ChronoGAM and those generated by DAGMM. Upon inspection, it is easier to differentiate the instances of the interest class. ChronoGAM representations are better at capturing temporal relationships in data,

allowing for the separation of instances based on their patterns over time, and may be beneficial for other tasks, which is also an advantage over IsolationForest. However, there are cases where a more precise adjustment is needed, such as in the FordA set, which was purposely brought to the grid to demonstrate that even when avoiding cases of numerical instability, some sets still have particularities that must be addressed, whether by adjusting filters and kernels or by using different metrics instead of cosine similarity to generate the final representations of the data.

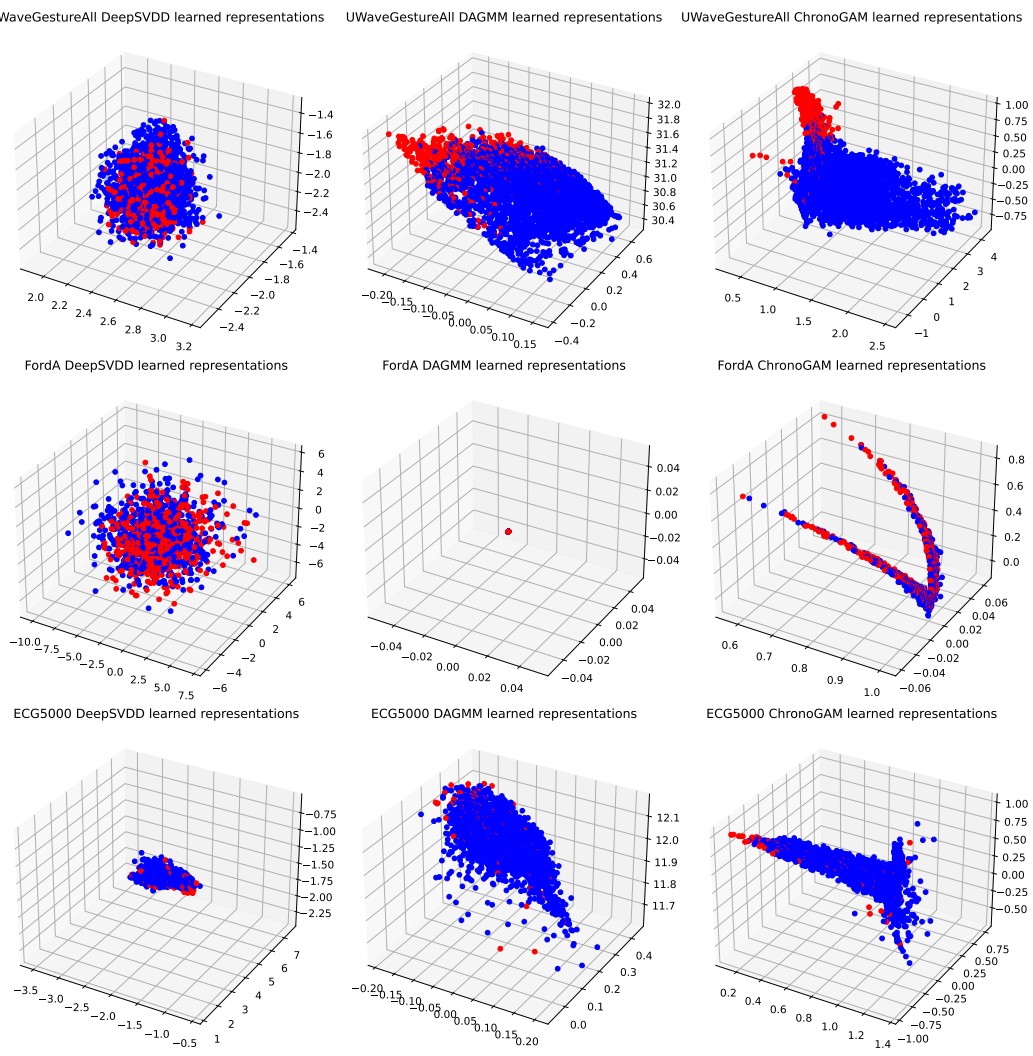

Figure 5: Representations learned through DeepSVDD, DAGMM and ChronoGAM. The sets used were UWaveGestureLibraryAll (interest class: 1), FordA (interest class: 1) and ECG5000 (interest class: 3). All representations generated were on test set instances never before seen by the methods. Interest instances are in blue and non-interest in red.

