# OpenReview forum: "ChronoGAM: An End-to-End One-Class Time Series Gaussian Mixture Model"
_ICLR.cc/2024/Conference — Submitted to ICLR 2024_

### Official Review · Reviewer_ahgt · 2023-10-31

**Soundness:** 2 fair
**Presentation:** 3 good
**Contribution:** 3 good
**Rating:** 6
**Confidence:** 3

**Summary:**

This paper proposes a method specifically targeted at time series data in the one-class learning problem. The proposed method consists of an autoencoder type with a convolutional layer that can capture the basic features of time series and a Gaussian mixture model that can capture multiple cluster structures in the latent space.
The authors have embedded some innovations in the proposed method to solve some problems that related methods have had in practical use. Specifically,
- The proposed method is a data-driven method using data energy instead of manual thresholding, which is often required in one-class learning.
- Instead of the ${\rm exp}$ function, which causes instability in the calculation of data energy, the proposed method improves the stability of the calculation by proposing an ${\rm explu}$ function.

Furthermore, the paper demonstrates the practical usefulness of the proposed method with objective measures for a remarkably large number of experimental data.

**Strengths:**

This paper proposes a very solid way to solve in a reasonable way several problems that have been practically and empirically intractable in the latest developments in one-class learning research. In particular, the following points can be listed as notable strengths:
- The paper is very well written, the arguments are coherent, and the organization is well thought out for a diverse audience. The authors' perspectives on the latest developments and current issues are well organized, especially in the context of one-class learning.
- The objective evaluation experiments that demonstrate the usefulness and effectiveness of the proposed method are notably large, comprehensive, and fair. It is suggested that the models assumed by the proposed method (time series representation by convolutional auto encoder and cluster structure in latent space by Gaussian mixture model) fit well on many real data.

**Weaknesses:**

My concern is that the discussion of the validity of the improvements from the Deep Autoencoding Gaussian Mixture Model (DAGMM) [Zong+2018], which is the inspiration for this study, may be somewhat lacking. The model used in this paper can be viewed as a legitimate successor to DAGMM from an overarching and conceptual perspective. Therefore, I was reading the manuscript with the expectation that the manuscript would carefully explain the validity of each improvement from it. Certainly, I agree that empirically and experimentally those improvements work well, as the authors have shown us in their evaluation experiments, but I am left with some points that are not clear to me why they work well from the perspective of scientific and technological development. My concerns can be summarized as follows.
- This paper introduces a convolutional approach to time series modeling with reference (maybe, as a standard method described in [Wang+2017]) to existing research on time series classification problems. It seems to me that the strengths of this approach (i.e., for what time series can well-fitted features be captured) and the weaknesses (what time series are difficult to represent) have not been adequately discussed.
- I am having difficulty understanding the validity of the explu function, one of the devices of the proposed method. explu function can be seen as an approximation/substitution for computational stability of what is originally an exp function, but I have not been able to find an explanation of what the sacrifice is.

These two concerns could be misunderstandings due to my lack of understanding. If there is any misunderstanding on my part, I would be very grateful if the authors could clear it up in their response.

**Questions:**

I would like to ask the following questions to see if my understanding of the two concerns I raised in the weaknesses section above is incorrect.

(1) Regarding the convolution approach that this paper employs to represent time series.
- Is the adoption of the convolution approach the recommended setting by the authors? Or is it adopted because it is known as the most basic and standard in the field concerned? For example, if the user has prior knowledge of the time series of interest, is it easy to change the proposed method to a time series model specific to that interest (in this context, the design of the AE layer)?
- Is this a robust setting for local (near temporal) features of the time series of interest, as the standard convolution assumes? Or is it also possible to capture very distant temporal dependencies (in the extreme case, where the first frame determines the last frame) or periodic structures?

(2) Regarding ${\rm explu}$ function.
I believe that instability in the computation of covariance (the inverse of covariance) is a challenge often faced in statistical machine learning. For example, this problem often arises in Gaussian process regression (GPR) when the observed data are of high dimension. In GPR, this problem is usually addressed indirectly (the direct motivation is to reduce the computational complexity of the inverse of the covariance matrix) using, for example, variational methods or induced points [Titsias+2009]. While these methods require rather complicated handling, I feel that the explu function in this paper is a very simple and impressive potential new way to deal with this problem. So let me ask a question.
- What are the disadvantages that arise when replacing the exp function with the explu function?
- Can that disadvantage be ignored in practical application situations?

M. Titsias. Variational learning of inducing variables in sparse Gaussian processes. In Proceedings of the Twelth International Conference on Artificial Intelligence and Statistics, pages 567–574, 2009.

---

> ### Author Response · Authors · 2023-11-21
>
> Thank you for the review. There are very interesting points to explore in your criticisms, which are, in a way, encouraging to answer. Starting with questions about the representation of time series:
>
> **“Is the adoption of the convolution approach the recommended setting by the authors? Or is it adopted because it is known as the most basic and standard in the field concerned? For example, if the user has prior knowledge of the time series of interest, is it easy to change the proposed method to a time series model specific to that interest (in this context, the design of the AE layer)?”**
>
> As we carried out extensive testing on different data sets generated and transformed to adapt to a One-Class Learning problem, we adopted an autoencoder structure capable of generalizing some information from different sets without major problems. Therefore, the proposed architecture, its layers, and parameters are a general recommendation that can be changed in different sets, changing the structure of the autoencoder and parameters such as the number of layers, filters, and size of the applied kernels. Activation is a recommendation given in the article [1], which presents a theorem about neural networks and the ability to reconstruct periodic patterns, so we adopted it to preserve the periodicity of the series without giving up other advantages.
>
> [1] Ziyin, Liu, Tilman Hartwig, and Masahito Ueda. "Neural networks fail to learn periodic functions and how to fix it." Advances in Neural Information Processing Systems 33 (2020): 1583-1594.
>
> **“Is this a robust setting for local (near temporal) features of the time series of interest, as the standard convolution assumes? Or is it also possible to capture very distant temporal dependencies (in the extreme case, where the first frame determines the last frame) or periodic structures?”**
>
> We evaluated the capacity of the model through the different sizes of the series available in the data sets, see Table 3. It is possible to observe that the performance is maintained for time series of up to 1000 observations, being impaired in series with a size greater than that or in series very small. Once this is observed, one of its causes is the size of the kernels applied during the convolutions. Note that optimizing different kernel sizes to find the best solution for each of the 652 generated datasets would result in an unfeasible experiment load. Therefore, we opted for a model with a generalist kernel size that obtains good performance for most solutions. However, this does not prevent other works from exploring these parameters to increasingly improve results in very large time series or treat this type of situation with other structures capable of capturing these long-distance dependencies.
>
>
> Regarding the Explu function:
>
> **“What are the disadvantages that arise when replacing the exp function with the explu function?”**
>
> By removing the positive exponential side of the function, we have a loss in the size of the penalty for positive values since the positive side of the function's domain becomes linear. Another point is that we have a constant derivative equal to 1 on this side of the function, which can bring some disadvantages in specific scenarios.
>
> **“Can that disadvantage be ignored in practical application situations?”**
>
> We adopt the “there is no free lunch” theorem in this case. It is almost impossible to try to specify the disadvantages in all domains of all problems. However, a brief investigation into the nature of the problem can give clues about the application of the explu.
>
> We can exemplify this with the case we used. In our scenario, the existence of instances with very high energy values could be maintained as follows:
>
> Replacing exp with explu we would have a constant derivative that would avoid exploding gradients due to high values.
> The model compensates in the layer weights for the need for high values for certain instances in the one-class scenario.
>
> Therefore, the disadvantage of explu could be kept under control in the domain, and by keeping small values close to 0 in the negative domain, we would still have a penalty applied to these cases.
>
> Assuming a scenario with an exp applied from a threshold and 0 for values below the threshold. We could somehow penalize values below the threshold and avoid the explosion of the gradient by replacing it with an explu, which would make learning slower in some cases due to the derivative being equal to 1. However, it would avoid instabilities caused by extremely large values when we apply the derivative of an exponential.

---

> > ### Comment · Reviewer_ahgt · 2023-11-23
> > **My two concerns have been resolved.**
> >
> > I greatly appreciate the author's thoughtful response. The content of the answer is very valid for me. My two concerns have been addressed. Therefore, I would like to raise my grade.

---

### Official Review · Reviewer_5RCZ · 2023-10-31

**Soundness:** 1 poor
**Presentation:** 2 fair
**Contribution:** 1 poor
**Rating:** 1
**Confidence:** 5

**Summary:**

This paper addresses one-class detection for time series. To this end, the authors use an auto-encoder similar to the one introduced by Zong et al. (2018), while including some variations to tackle the temporal features as proposed by Fawaz et al. (2019).

**Strengths:**

This paper tackles an interesting problem, which is one-class detection on time series with deep learning.

**Weaknesses:**

A major issue of this paper is the incremental contribution. The authors use roughly the same the auto-encoder proposed by Zong et al. (2018), which was introduced and investigated to address unsupervised anomaly detection using a Gaussian mixture model. The only difference seems to be the convolution layers as proposed by Fawaz et al. (2019), and the snake activation of Ziyin et al. (2020).

Another major issue is that the paper does not demonstrate clearly the relevance of these modifications. The authors should conduct an ablation study in order to show how the implemented modifications impact the obtained results, including convolution layers of Fawaz et al. (2019) and the snake activation of Ziyin et al. (2020).

Another major issue is experiments and comparative analysis. The authors compare the proposed method to 4 other detection methods: OCSVM, IsolationForest, DeepSVDD and DAGMM. All these methods are not relevant to address time series. Therefore, this is not enough as the authors did not provide any comparative analysis with related methods, namely methods from the deep leaning literature that address anomaly detection in time series. For a review, see
* Choi, Kukjin, Jihun Yi, Changhwa Park, and Sungroh Yoon. "Deep learning for anomaly detection in time-series data: review, analysis, and guidelines." IEEE Access 9 (2021): 120043-120065.
See also related methods
* Kim, Siwon, Kukjin Choi, Hyun-Soo Choi, Byunghan Lee, and Sungroh Yoon. "Towards a rigorous evaluation of time-series anomaly detection." In Proceedings of the AAAI Conference on Artificial Intelligence, vol. 36, no. 7, pp. 7194-7201. 2022.
* Xu, Jiehui, Haixu Wu, Jianmin Wang, and Mingsheng Long. "Anomaly transformer: Time series anomaly detection with association discrepancy." arXiv preprint arXiv:2110.02642 (2021).
* Zhang, Chuxu, Dongjin Song, Yuncong Chen, Xinyang Feng, Cristian Lumezanu, Wei Cheng, Jingchao Ni, Bo Zong, Haifeng Chen, and Nitesh V. Chawla. "A deep neural network for unsupervised anomaly detection and diagnosis in multivariate time series data." In Proceedings of the AAAI conference on artificial intelligence, vol. 33, no. 01, pp. 1409-1416. 2019.
* Garg, Astha, Wenyu Zhang, Jules Samaran, Ramasamy Savitha, and Chuan-Sheng Foo. "An evaluation of anomaly detection and diagnosis in multivariate time series." IEEE Transactions on Neural Networks and Learning Systems 33, no. 6 (2021): 2508-2517.
* Li, Dan, Dacheng Chen, Baihong Jin, Lei Shi, Jonathan Goh, and See-Kiong Ng. "MAD-GAN: Multivariate anomaly detection for time series data with generative adversarial networks." In International conference on artificial neural networks, pp. 703-716. Cham: Springer International Publishing, 2019.
* Zhang, Yuxin, Yiqiang Chen, Jindong Wang, and Zhiwen Pan. "Unsupervised deep anomaly detection for multi-sensor time-series signals." IEEE Transactions on Knowledge and Data Engineering (2021).
* Tuli, Shreshth, Giuliano Casale, and Nicholas R. Jennings. "Tranad: Deep transformer networks for anomaly detection in multivariate time series data." arXiv preprint arXiv:2201.07284 (2022).
* Carmona, Chris U., François-Xavier Aubet, Valentin Flunkert, and Jan Gasthaus. "Neural contextual anomaly detection for time series." arXiv preprint arXiv:2107.07702 (2021).

Finally, there are some spelling and grammatical errors, such as “autoncoder”, “cossine similarity function”, “recomendation”.

**Questions:**

Why didn't you compare to other deep anomaly methods for time series ?

Why there is no ablation study that allows to demonstrate the relevance of the proposed modifications ?

---

> ### Author Response · Authors · 2023-11-21
>
> Firstly, we would like to thank you for the review and we will try to resolve the points that remained as doubts. To do this, let's contextualize some points:
>
> One-class learning is a type of supervised learning that involves modeling examples belonging to a single class. According to the definition, only instances of the class of interest are available for training. On the other hand, Anomaly Detection is a type of learning that encompasses a set of techniques that can be supervised, semi-supervised, or unsupervised learning (training with normal and anomaly classes, i.e., using binary learning that is different from one-class learning). Anomaly Detection aims to identify abnormal instances or points in a data set and can also be referred to as Outlier Detection. [1]
>
> With these definitions, we can move on to the characteristics that differentiate both:
>
> - Nature of the datasets: in some cases of outlier detection, there are anomalies present during the training set. In time series, differences like the data sets become more noticeable, while anomaly detection or outlier detection can focus on capturing anomalous observations that may exist in the same class, one-class is a classification problem, where an instance (entire series) will or will not belong to a class of interest. This way, Anomaly Detection can handle point-wise detection.
> - Characteristics of the problems: One of the main differences lies in the characteristics of the problem definition. In One-Class Learning, we only have instances of the class of interest in the data set, and this statement makes it a supervised learning problem, since the presence of unlabeled instances or instances outside this class directly impacts the learning and performance of the model, changing the paradigm of the task performed. In Anomaly Detection, some works focus on different statements, such as the mandatory presence of anomalous instances in the training set, the mandatory non-existence of these instances, or even semi-supervised learning more similar to Positive and Unlabeled Learning (PUL), where normal instances are present next to unlabeled ones.
> - Structure of the models: Due to the characteristics and statements, the models have different learning strategies, using binary objective functions or those that collapse when brought to one-class scenarios. That said, checking these models and studying and adapting them to work in one-class learning scenarios requires a greater effort, which would involve different work.
>
> Therefore, one-class learning methods cannot be directly applied to anomaly detection or vice versa. [1, 2]
>
> Another important point highlighted by researchers such as Eamonn Keogh is that papers that work with anomaly detection suffer from several problems, from the nature of the datasets that suffer from several flaws to complex methods solving simple problems. [3]
>
> [1] Perera, Pramuditha, Poojan Oza, and Vishal M. Patel. "One-class classification: A survey." arXiv preprint arXiv:2101.03064 (2021).
>
> [2] Seliya, Naeem, Azadeh Abdollah Zadeh, and Taghi M. Khoshgoftaar. "A literature review on one-class classification and its potential applications in big data." Journal of Big Data 8.1 (2021): 1-31.
>
> [3] Eamonn Keogh. Irrational Exuberance Why we should not believe 95% of papers on Time Series Anomaly Detection. Available at: https://youtu.be/Vg1p3DouX8w?si=_NvqMdF9Ir-DTMnl (2021).
>
> **“Why there is no ablation study that allows to demonstrate the relevance of the proposed modifications ?”**
>
> We included a more in-depth study of the results in the appendices. However, due to limited space, we focused on the main points during the analysis of the results in the paper. We show and compare the approach using the new proposed loss function and without using it (adapting only the autoencoder) in Figure 2 (CDD).
>
> We also raised some analyses about the sets' characteristics and the autoencoder's performance in capturing temporal dependencies. To see the relevance of these results, simply compare ChronoGAM (wo regularizer) in Figure 2 with DAGMM, which implements the same loss function and differs in the autoencoder. The gain shows that adapting the model structures to work with temporal dependencies is enough to surpass classic Machine Learning models. However, it is still unable to perform without a significant statistical difference when we analyze it together with the complete ChornoGAM, which implements the new loss function.

---

> > ### Comment · Reviewer_5RCZ · 2023-11-22
> > **Acknowledgments**
> >
> > We thank the authors for their reply. However, the issues that I have raised are not addressed.
> >
> > Of particular interest is why the authors did not compare the proposed method to other deep anomaly methods that have demonstrated their relevance on time series.
> >
> > The reply of the authors in this rebuttal is that papers that work with anomaly detection suffer from several problems, citing Eamonn Keogh presentation "Irrational Exuberance Why we should not believe 95% of papers on Time Series Anomaly Detection". This reply raises other questions, if the authors are following thus presentation:
> > - Please consider that this is only a presentation, and not a rigorous scientific analysis: 95% of papers is just an illustrative figure, and the presentation is only about a couple of papers.
> > - That presentation provides a benchmark that aims to overcome the raised issues. The authors of the submitted paper did not investigate this benchmark.
> > - The presentation has a positive takehome message, including several constructive points following "We should see these facts as a wonderful opportunity..."
> > At the end, the authors of the submitted paper consider that such a presentation invites the researchers not to compare to other methods from the literature. It is a pity for advancing research.
> > Please check my review for related literature on anomaly detection on time series.
> >
> > Another point in the rebuttal is the sentence "one-class learning methods cannot be directly applied to anomaly detection or vice versa. [1, 2]".
> > I did not understand what the authors want to say, since, in the contrary, there are too many one-class learning methods applied to anomaly detection and vice versa, as provided by the surveys [1, 2].
> > I hope that the authors can acknowledge that their submitted paper is not the first in the literature to apply one-class learning methods applied to anomaly detection.
> >
> > The issue of the ablation was raised by several reviewers. The reply of the authors is not correct because the Appendix does not contain an ablation study and no interesting information or any deep analysis.
> >
> > Moreover, I did not understand why 2/3 of the rebuttal is a general presentation of anomaly detection. To the best of my knowledge, and considering my review, I think that I know what anomaly detection is.

---

> > > ### Author Response · Authors · 2023-11-22
> > >
> > > Thanks again for the response. We understand that you know what Anomaly Detection is, and we apologize for any misunderstanding, the idea of contextualizing it was to explain the differences between one-class learning and anomaly detection, as well as use what Eamonn says in his presentation to exemplify the existing problems in anomaly detection and which do not affect the area of one-class learning. That's why we tried to define both, especially the concept of one-class in the answer above, making the differences clear.
> > >
> > > Regarding the point of appendices, the text was not clear. However, the correct item would be "We will add an appendix with these discussions". The idea is to add the appendix to strengthen these points discussed, and we are glad you highlighted them as it works to clarify these differences.

---

### Official Review · Reviewer_JBii · 2023-11-06

**Soundness:** 2 fair
**Presentation:** 2 fair
**Contribution:** 2 fair
**Rating:** 5
**Confidence:** 4

**Summary:**

This paper proposes an end-to-end one-class time series Gaussian mixture model to resolve the collapse of hyperspheres, manual thresholds, numerical instabilities, and even the use of unlabeled instances during training.

The proposed ChronoGAM aims at improving the temporal importance of the representations learned by the autoencoding system.

The techniques are to penalize the small values on the covariance matrix without resulting in exploding gradient propagation, causing numerical instabilities, and adapting the energy calculus to avoid the use of exponential functions.

Experimental results show the effectiveness.

**Strengths:**

- Simple and efficient reminiscences of numerical challenges in One-Class-Learning.
- Extensive experimental results show the effectiveness of the proposed method.

**Weaknesses:**

- Limited concrete evidence for how the numerical challenges in One-Class-Learning are released
- By using an autoencoder, the numerical problems are resolved but with limited theoretical justification - and no ablation study is presented.

**Questions:**

it is overall interesting to see that with an end-to-end design, the performance is enhanced. However, it can be improved via the ablation perspective of the experiments in such an unsupervised setting.

The impact of adopting an autoencoder is welcomed but it requires concrete examples and evidence regarding:
- what temporal and structural features are well captured.
- how it helped in resolving the numerical challenges and why an autoencoder is the solution in that situation.

---

> ### Author Response · Authors · 2023-11-21
>
> Thank you for the reviews and pointed questions. Based on the demonstration of the problems found in (Ruff et al., 2018), we inserted an appendix to highlight the numerical challenges that involve the use of hyper-sphere loss, including the n-ball problem [2], as well as the experimental results that highlight the numerical instability of the DAGMM in Figure 3.
>
> The problems of numerical instability were solved by inserting a regularizing term of the covariance matrix different from that proposed in DAGMM, see the final part of Equation 4, as well as equations 2 and 3 that insert the use of the proposed Explu function.
>
> “what temporal and structural features are well captured.”
>
> Unlike a simple MLP network, a convolutional filter can be used to obtain results for all timestamps of a time series, allowing the learning of invariant filters along the temporal dimension. These CNN networks allow the learning of these filters dynamically, optimizing the parameters for each data set used and learning filters that make it easy to discriminate between classes [1].
>
> See in Figures 4 and 5 that this type of network optimizes the generation of more discernible embeddings between time series instances, making the representations more robust for the method.
>
> [1] Ismail Fawaz, Hassan, et al. "Deep learning for time series classification: a review." Data mining and knowledge discovery 33.4 (2019): 917-963.
>
> [2] Weisstein, Eric W. "Hypersphere." From MathWorld--A Wolfram Web Resource. https://mathworld.wolfram.com/Hypersphere.html
>
> “how it helped in resolving the numerical challenges and why an autoencoder is the solution in that situation”
>
> In this case, the method can work more robustly with better embeddings. However, in the results, we show that just adapting the autoencoder will not solve the problem of numerical instability. The method started from this hypothesis to modify the loss function and add the term:
>
> $\text{min} \left ( \sum^K_{k=1}\sum^d_{j=1}-\log(\mathbf{\hat{\Sigma}}_{kjj}), 100 \right )$
>
> In Figure 2, it is possible to observe the CDD constructed from the results obtained, so we have an implementation that only adapts the autoencoder (ChronoGAM wo Regularizer) and the complete implementation of the method (ChronoGAM) that uses the new proposed loss function. In this Figure, it is possible to observe that we can obtain better results by adapting the method with the autoencoder and using the new loss function.

---

### Official Review · Reviewer_yKkP · 2023-11-09

**Soundness:** 2 fair
**Presentation:** 2 fair
**Contribution:** 2 fair
**Rating:** 3
**Confidence:** 4

**Summary:**

The paper is focused on the problem of one-class learning with time series data. Existing methods face problems like hypersphere collapse and numerical instability, etc. The proposed method uses a Gaussian Mixture Model, combines an autoencoder for feature extraction, and improves temporal importance. It's tested on many datasets and shows outperformance in 369 of them.

**Strengths:**

This is an applied paper and has little technical contribution.

The experiment is relatively comprehensive on 85 benchmark datasets and derived 652 datasets.

**Weaknesses:**

1. The author claims the gain of the proposed method in 369 datasets out of a total of 652. The issue with this massive test is the lack of insights into the reasons for the proposed method's effectiveness and ineffectiveness on these many datasets.

2. The proposed method appears to have no significant technical novelty; thus, it is a question of where the outperformance stems from. The authors are expected to provide a conceptual or theoretical explanation of why the proposed method can outperform in addition to experimental results.

**Questions:**

See the weakness section above.

---

> ### Author Response · Authors · 2023-11-21
>
> Firstly, we would like to thank you for the review and the suggestions. We will try to resolve as many of them as possible below to improve the quality and contributions of the paper.
>
> We modified the discussion of the results to clarify discussions of why ChronoGAM (the proposed method) outperforms other methods.
>
> “(...) Unlike traditional machine learning methods, such as IsolationForest and OCSVM, which interpret observations as attributes without temporal dependence, ChronoGAM uses convolutions to capture this dependence and thus obtains better results. Compared to its competitors that use deep learning strategies, DeepSVDD suffers from the problem of hypersphere collapse, leading to poor results which ChronoGAM easily overcomes due to not using a hypersphere loss.
>
> ChronoGAM has been more successful than its main competitor (DAGMM) for two primary reasons. Firstly, its autoencoder architecture is more flexible, using convolutional structures that can detect temporal correlations in the data, as demonstrated in various studies on time series classification (Fawaz et al, 2019, Wang et al, 2017, Zong et al, 2018). In contrast, DAGMM only uses an MLP-based autoencoder, which cannot capture these dependencies. The second reason is the covariance matrix regularizer, while the DAGMM uses a
>
> $p=\sum^K_{k=1}\sum^d_{j=1}\frac{1}{\mathbf{\hat {\Sigma}}_{kjj}}$
>
> which can easily cause very large values, making it difficult to correctly propagate the error to the task, leading to numerical instabilities and exploding gradients, ChronoGAM uses the penalty $p=\text{min} (\sum^K_{k=1}\sum^d_{j=1}-\log(\mathbf{\hat{\Sigma}}_{kjj}), \alpha)$ where $\alpha = 100$ , so values that would cause the gradient to explode and cause numerical instabilities as well as disregarding the other tasks that make up the objective function to simply minimize the covariance matrix are more easily avoided. (...)”
>
> We emphasize that the innovation of the proposed method lies in the new loss function used, proposing the explu to avoid very high values as occur in the exp, in addition to the motivation of the term that will regularize the covariance matrix that is used by the method (avoiding division to extremely low values that causes the exploding gradients problem). As for the results, the detailed results of each dataset, label, and metric will be made available in the project repository. To clarify this issue, we modify the following sentence:
>
> “(...) The experimental evaluation codes and results are publicly available1. (...)”
>
> And later the link will be updated to the project's public repository (link omitted for double-blind purposes).

---

### Author Response · Authors · 2023-11-21

We would like to thank the reviewers for their points and suggestions. Reviewing an article is always a work for the benefit of the scientific community. We tried to clarify the points in each of the specific answers and indications of changes and inclusions in the appendices, in general, we tried to clarify and resolve doubts with the material already existing in the article without making major changes, in cases where the need for inclusion was noted of material, we chose to include the discussions in the appendices due to limited space.

---

### Meta-Review · Area_Chair_UHop · 2023-12-05

**Metareview:**

This paper considers the problem of one-class detection in time-series domain. The proposed method consists of an auto-encoder type model to capture temporal features and a Gaussian mixture model that can capture multiple cluster structures in the latent space.

The reviewers' agree that the technical innovation is incremental, missed important related work, lacks ablation results to explain the improvements and comparison with deep anomaly detection for time-series domain. Therefore, I recommend rejecting the paper and encourage the authors' to revise the paper based on the review comments for re-submission.

**Justification For Why Not Higher Score:**

Lot of weaknesses as mentioned in the meta review.

**Justification For Why Not Lower Score:**

N/A

---

### Decision · Program_Chairs · 2024-01-16

Reject